# Fear and Attitude towards SARS-CoV-2 (COVID-19) Infection in Spanish Population during the Period of Confinement

**DOI:** 10.3390/ijerph19020834

**Published:** 2022-01-12

**Authors:** Ana María Recio-Vivas, Isabel Font-Jiménez, José Miguel Mansilla-Domínguez, Angel Belzunegui-Eraso, David Díaz-Pérez, Laura Lorenzo-Allegue, David Peña-Otero

**Affiliations:** 1Department of Nursing, Faculty of Biomedical and Health Science, Universidad Europea de Madrid, 28670 Madrid, Spain; anamaria.recio@universidadeuropea.es (A.M.R.-V.); isabel.font@universidadeuropea.es (I.F.-J.); laura.lorenzo@universidadeuropea.es (L.L.-A.); 2Medical Anthropology Research Centre, Department of Quantitative Methods at the Faculty of Nursing, Rovira i Virgili University, 43002 Tarragona, Spain; angel.belzunegui@urv.cat; 3Respiratory Nursing Department at SEPAR, Respiratory Nurse at the Pneumology and Thoracic Surgery Service of the Hospital Universitario Nuestra Señora de Candelaria (Tenerife), 38010 Santa Cruz, Spain; davdiaper82@gmail.com; 4Respiratory Nursing Department at SEPAR, Nurse Member of the IDIVAL and IiSGM Research Institutes, 28007 Madrid, Spain; david.penao@scsalud.es; 5Hospital de Sierrallana, Cantabrian Health Service, 39300 Torrelavega, Spain

**Keywords:** SARS-CoV-2, coronavirus infections, pandemics, fear, public health, preventive health services

## Abstract

In January 2020, the WHO classified SARS-CoV-2 infection as a public health emergency and it was declared a pandemic on 11 March 2020. The media warned about the danger of infection, fuelling the population’s fear of the new situation and increasing the perception of risk. This fear can cause behaviour that will determine the course of the pandemic and, therefore, the purpose of this study was to analyse the fear of infection from COVID-19 among the Spanish population during the state of emergency. A cross-sectional, descriptive observational study was conducted with 16,372 participants. Data on sociodemographic factors, health factors, risk perception and fear were collected through an online survey. Level of fear is associated with older age, a lower level of education, having a person infected with SARS-CoV-2 in the immediate surroundings and living with and belonging to the most socioeconomically vulnerable group of people. Risk perception is associated with increased preventive behaviour. This paper provides relevant information for the public health sector since it contributes first-hand knowledge of population data that is highly useful in terms of prevention. Understanding the experiences of people in this pandemic helps to create more effective future intervention strategies in terms of planning and management for crisis situations.

## 1. Introduction

In January 2020, the World Health Organization (WHO) classified the coronavirus infection, known as SARS-CoV-2, as a public health emergency [1]. It started in Hubei Province, China, in December 2019 [2] and was declared a pandemic on 11 March 2020.

This new disease, known as COVID-19, is transmitted from person to person primarily through respiratory droplets, although other forms of infection have been documented, such as faecal–oral transmission and through fomites. COVID-19 incubation period ranges from 2 to 14 days, the median is at 5.1 days and it has been determined that 97% of infected subjects will develop symptoms within 11.5 days [3].

The symptoms vary, but the most common include fever, cough, sore throat and general malaise [2]. However, some people are asymptomatic while others develop a serious respiratory condition, and the lack of a rapid diagnostic test forced clinics to set standard measures based on clinical suspicion [4].

The number of reported cases in the European region was clearly on the rise, according to data provided by the WHO European Region in its report for week 12 (16th–22nd of March 2020). They also reported that the most vulnerable section of the population were people over 60 and that 1 in 10 of those infected were healthcare workers. On the positive side, 87% of the people infected have recovered. In Spain, during the first half of March 2020 (the weeks prior to our study), 182.816 confirmed cases of COVID-19 had been reported, with a total of 60,000 hospitalised patients, 5000 of them in ICU, and 19,130 deaths, with a mean of 800 deaths per day. The age group with the highest number of deaths was the over-70s, accounting for more than 80% of deaths in both men and women [5].

Faced with this situation, different countries around the world advised people to take a series of preventive actions, such as hand washing and social distancing, in order to protect themselves and stop the spread of the disease [1]. These measures have been endorsed by the population as shown in the McFadden study [6], in which US population who support certain strict control measures (use of masks, staying at home, quarantines and travel restrictions) were surveyed.

In our country, the State of Emergency declared on 14 March 2020 forced the Spanish population to be confined to their homes: They were only allowed to leave for health reasons, to acquire food and essential goods and to go to the workplace just in case of persons working in so-called essential services (Royal Decree 463/2020 of 14 March).

### 1.1. Fear and Attitude towards SARS-CoV-2

Fear is an emotion provoked by the perception of impending danger, which may be real or imagined, may be experienced in the present or expected in the future. Fear arises from the aversion of both humans and animals to threat or risk and causes an immediate alarm reaction in the body that triggers a series of physiological changes [7].

Collective fear is defined as a shared fear by a large part of a group or society. The emergence of COVID-19, its meteoric expansion, and the large number of deaths among older adults, especially in Spain, has brought with it what is known as “collective fear”. This fear is provoked by the threat of an imminent danger, the SARS-CoV-2. Collective fear must be contained because if taken to its most irrational extremes, can become a disintegrating social factor [7].

When calibrated to detect real threats, fear acts as an adaptive response which channels the required energy to face such a potential threat. An inadequately calibrated fear will entail individual and social consequences, such as impulsive buying and other excessive behaviours, as seen during this health crisis, with supermarket shelves emptied because of stockpiling by the population. However, insufficient fear may cause people to ignore government regulations to slow the spread of coronavirus, it and may also lead to the implementation of reckless policies with disregard of the risks [8].

The media warned about the dangers of infection, fuelling the population’s fear of the new situation along with their perception of risk [9].

The rapid spread of the virus and its international impact amplified fears that the disease affects everyone, regardless of gender and other sociodemographic variables [10].

Knowing the perception of fear in population is necessary to understand to what extent it can determine the adoption of preventive measures against COVID-19 [11].

### 1.2. Preventive Measures

Social distancing and other preventive measures are actions that must be taken individually but also collectively, especially by appealing to people who are at low risk of contagion but who could infect others who are more vulnerable [12]. Currently there are no available vaccines in all countries, neither to all age groups. Additionally, some studies highlight the decrease of the immune protection offered by the vaccine over time. These facts support the need to maintain some essential protective measures, such as social distancing and the use of face masks, among others [13,14].

It should be noted that confinement measures can have adverse effects: some authors have revealed an increase in gender-based violence [15,16], addictions [17] and anxiety [18,19], thus altering individual well-being [20]. Some people do not go to hospitals for fear of infection, and some specialists have reported a rise in deaths due this phenomenon [21,22]. There is a fear among the population of losing their jobs and not receiving their salaries [23], especially those who cannot work from home [24].

These prevention measures have been tested at other times throughout history, such as during the Spanish flu pandemic in 1918 and more recently the SARS-CoV outbreak in 2002 or the H5N1 2009 virus alert. In all these situations, a great number of public health preventive measures were promoted to be complied by population and so the virus effects were mitigated [25]. It must be noted that adequate information aimed at the population lead to greater preventive behaviour and helps to reduce fear of the possible effects of the current SARS-CoV-2 pandemic [26].

Risk perception and fear of infection correlate with a stricter compliance with rules and prevention measures [27]. In April 2020, Spain had been confined for more than 4 weeks, and there was no clear date set for when confinement would end, so we wondered what the population was concerned about; was there a sense of fear? Which part of the population was most likely to be afraid? Was there a correlation between perceived fear and compliance with the imposed rules? In order to answer these questions, the aim of this study was to analyse the fear of infection from COVID-19 among the Spanish population during the State of Emergency declared by Royal Decree on 14 March 2020. We consider that exploring how this threat is perceived by the population helps to identify some of the negative individual and social consequences of the coronavirus pandemic. Understanding correlation between fear and adherence to norms could help public health services to establish significant predictors of population behaviour and compliance with rules [8], and to support the population in complying with imposed preventive measures [28]. By combining these significant predictors makes it possible to elucidate different scenarios on which to design preventive policies.

## 2. Methods

### 2.1. Study Design and Participant Selection

A cross-sectional descriptive observational study was carried out which involved a total of 16,372 residents in the Autonomous Communities of Madrid, Cantabria and the Canary Islands. The choice of the sample is intended to represent three types of population characteristic of the country, namely: a mainly urban environment (Madrid, Spain), a rural environment (Cantabria, Spain) and an island environment (Canary Islands, Spain).

Participants who did not meet the eligibility criteria were excluded from the study, specifically, those who did not give informed consent, did not fill out the survey completely, or were under the age of 18. Finally, a total of 16,201 people (98.9% of participants) were included in this study, 71.1% of which were women (n = 11,521). The average age was 46 years old (SD 12.58).

Since the objective was to collect as many responses as possible, a consecutive nonprobability (snowball) sampling technique was used. As this is not a random sample, but rather a snowball sampling where the final sample is generated through an accumulative effect, and given that the sample obtained in this study was rather large, it can be viewed as conditionally representative within each sample subgroup [29]. In our case, to correct for sample bias, poststratification was used to adjust the rough estimates. Along these lines, Wang, Rothschild, Goel and Gelman show that, for election polling, and with appropriate statistical adjustments, polls with nonrandom samples can be used to generate precise results, and this oftentimes can be achieved faster and at a lower cost than traditional polling methods. These same authors conclude that nonrepresentative polling is promising, not only for predicting the winner of an election, but also for measuring public opinion on a wide range of social, economic and cultural issues [30]. According to these authors, when there is a need to identify and trace crucial events that affect public opinion, nonrepresentative polling offers the possibility of profitable ongoing data collection. Nevertheless, standard representative polling will continue to be an essential tool.

The data collection process was carried out over five days from the 16th to the 21th of April 2020 during the period of mandatory confinement enacted following the declaration of the State of Emergency in Spain. Data were collected through an online survey distributed via email and social networks of the researchers and participating institutions (WhatsApp, Instagram, Facebook and Twitter). In turn, the collaboration of the population was requested through communiqués and publications on the websites of healthcare institutions and organisations: the official nursing associations of Madrid, Cantabria and the Canary Islands; the General Council of Nursing of Spain; the network of the Subdirectorate of Care of the Cantabrian Health Service; the SATSE Nurses’ Union and other networks of scientific societies of nursing and medicine (such as Spanish Society of Pneumology and Thoracic Surgery, SEPAR). In addition, notifications were sent via the Cantabria Health APP.

### 2.2. Tool and Study Variables

The survey had 59 items divided into 4 content areas: sociodemographic factors, characteristics of the experience, health factors and risk perception. The free tool “Google Form” was used to craft the survey. The first proposal was evaluated by the research team, which has experience in the healthcare, research and teaching fields. The survey also included an explanation of the project, requirements for participation, instructions for compliance and acknowledgement, informed consent and mandatory questions.

The questionnaire was developed and sent to participants in Spanish, the official language throughout Spain. In the first lines of the survey, respondents were asked to answer the survey if they lived in Madrid, Cantabria or the Canary Islands. In addition, a question referring to nationality was included to determine the characteristics of the sample in more detail.

### 2.3. Analysis

Fear of infection was defined as a dependent variable. To perform a subsequent binary logistic regression analysis, this variable included two response categories (Yes/No). This categorisation was based on the lack of a commonly accepted instrument to measure fear of infection. In this regard, Collins’ notes that in the literature prior to 2020 there are few quantitative measures of fear related to infection [31]. During the most recent comparable epidemic, the 2002–2004 SARS epidemic, several measures arising from ethnographic methods were used. As independent variables, the following were selected: age, gender, marital status, level of education, employment situation, active population (working-age people who are in paid employment or in search of paid employment), annual salary, perception of the family financial situation, cohabitation, health status during confinement, having a healthcare worker in the family, infections in the home, fear of a relative becoming infected, means of obtaining information (WhatsApp, social networks, official sources or scientific sources) and protection measures used (use of face mask and gloves when leaving home). The binary logistic regression included “fear of infection” as a dependent variable, coded as 0 = No and 1 = Yes. Independent variables included gender (0 = Male; 1 = Female); age (continuous); level of education (0 = Elementary; 1 = Secondary; 2 = University); cohabitation (0 = Lives alone; 1 = Lives with others); health status during the pandemic (0 = Good, Very good; 1 = Average; 2 = Poor, Very Poor); perception of family financial situation during the pandemic (0 = Good; 1 = Average; 2 = Poor); personal protection measures used (0 = Adequate; 1 = Inadequate) and infection (number of people, including the respondent, who had COVID-19 symptoms in the household).

We started by calculating the percentages of the variables later considered for predicting the fear of infection and testing variable independence with chi-square. The Odds ratio (OR) was also calculated to know the probability of suffering from fear associated with the variable categories that would be included in the logistic regression analysis. The effect size was calculated with Cohen’s *d* [32].

A binary logistic regression analysis was performed with IBM SPSS v.21 for the predictive analysis of fear of infection. The variables included in the logistic model were gender, age, level of education, infection of a relative or someone close to them, whether the subject lives alone or accompanied, the current state of perceived health, the family financial situation and the use of proper means of protection or lack thereof. Logistic regression is particularly suitable for assigning probabilities of occurrence of a phenomenon among individuals from the combination of categorical and quantitative variables. Moreover, the probabilistic results of this type of analysis associated with individuals help to identify groups or clusters and to develop prevention policies specifically aimed at these groups.

The analysis resulted in a new variable with the probabilities of risk, which was later used in mean comparison analyses for independent groups and ANOVA to observe differences in the different perception of fear of infection based on certain variables. The significance analysis of the *t*-tests and *F*-tests was completed by calculating the effect size using Cohen’s *d* [33].

All analyses were performed with the SPSS Statistics program IBM SPSS Statistics for Windows, version 26.0. (IBM Corp., Armonk, NY, USA).

## 3. Ethical Aspects

The study was carried out in accordance with the principles of the Declaration of Helsinki and the laws and regulations in force in Europe and Spain and was approved by the Drug Research Ethics Committee of Cantabria (code: 2020.159).

Given the exceptional circumstances of the pandemic and following indications from the European Medicines Agency and the Ministry of Health, Consumer Affairs and Social Welfare, written informed consent was requested at the start of the online survey. Users had to accept it to continue with the survey.

## 4. Results

The descriptive analysis of the study variables is detailed in Table 1.

All variables included in the regression equation present a statistically significant association with the variable “Are you afraid of becoming infected?”, except the variable “Having a healthcare worker family member” *(p =* 0.330), as can be seen in Table 2.

The objective of the regression analysis was to formulate a predictive equation of fear from the independent variables mentioned. The enter method was used after checking that the forward conditional and backward conditional methods produced the same results in terms of goodness of fit and predictive potential.

The analysis was performed on n = 13,786 (85.1%), leaving out 2415 cases (14.9%). The results of the analysis show a correct predictive value, 71.4%, much better at predicting people who perceive more fear than those who do not (with a cutoff value of 0.050). The omnibus test of model coefficients indicates the suitability of the model formulated with a *p* < 0.001 and a chi-square = 691.37. The goodness of fit with the Hosmer–Lemeshow test yields a value of 7.849 and a significance of *p =* 0.448, indicating that the model variables fit well.

The table with the variables included in the regression analysis and their coefficients shows that, except for the variable Current Health Poor, there is statistical significance for all *(p <* 0.01 or *p <* 0.05), with a confidence level of 95%. However, this variable was included in the regression equation to safeguard the substantive consistency of the categories of the input variable, as indicated by Hair [34].
P(Y = 1) = 1/1 + e^−(−1.07+0.36·Sex+0.69·PS+0.148·SS+0.293·COVID+0.124·Ffa+0.179·Ffb+0.332·Co+0.359·Cha+0.013·Age+0.66·AP)^(1)

As can be seen in Table 3, the probability of fear increases in women, in people with primary studies, if the family financial situation has worsened, in people with average health, in those living with other people, and if adequate means of protection are used. Furthermore, as age increases, the likelihood of fear of infection rises; similarly, fear increases if the person has been infected by COVID-19 or if a family member, friend or coworker has been infected (or has died). These variables can be considered risk factors, as they were already linked to fear in Table 2 in the first bivariate analysis.

Similarly, the ORs (Exp(β)) in the table indicate a higher probability of perceiving fear of infection in all variables included in the model (except for Current Health Poor).

From here, a comparison of means analysis for independent samples was carried out, comparing means in different sample subgroups using the *t* statistic, or the *F* statistic for ANOVA. The results are shown in Table 4.

As shown in Table 4, women are more likely to fear infection (0.728) as opposed to men (0.654) (the size effect measured by Cohen’s *d* is *d_Cohen_* = 0.765). People with a higher level of education are less likely (0.683) than those with basic education (0.824) (Cohen’s *d* for ANOVA is *d_Cohen_* = 0.564). Differences can also be seen between the active population (0.698) and the inactive (0.725) *(d_Cohen_ =* 0.266). People who live with others (0.715) are more afraid of being infected than those who live alone (0.645) *(d_Cohen_* = 0.70). Differences were found between those who live alone and those who do not live alone, as the number of household members is irrelevant; there is no distinction between households with 2, 3, 4 or more persons. However, widowed people (0.754) show a greater probability of fear than single, married and separated people. There are no significant differences between married and separated/divorced people (*p* = 0.756) (*d_Cohenv_* = 0.86). People who protect themselves adequately (0.753) have a higher probability of fear than those who do not (0.588) *(d_Cohen_ =* 2.33). There are also significant differences between those who perceive their household financial situation as good during the pandemic (0.684) and those who consider it to be average (0.722) or poor (0.746) *(d_Cohen_ =* 0.248). In terms of how people perceive their health during confinement, those who believe it has worsened and is now average (0.779) are more likely to perceive fear *(d_Cohen_* = 0.372). The fact of having a family member who is a healthcare worker does not increase the probability of perceiving fear. However, having a family member diagnosed with COVID-19 (0.731) does increase fear when compared to those who do not (0.706) *(d_Cohen_* = 0.313). Similarly, fear of a family member becoming infected increases the likelihood of fear of infection *(t =* −5.481; *p* < 0.001; (*d_Cohen_* = 0.238). Feeling at risk of illness is associated with the probability of being afraid: low (0.691), medium (0.712) and high (0.732) risk differ significantly, as shown in Table 3
*(d_Cohen_* = 0.402). The means chosen to learn about the pandemic also does not influence the probability of having a greater or lesser level of fear of infection (test *F* = 2.261; *p* = 0.104). 

Other results indicate that, as older people become more affluent, they are less likely to fear infection (see Figure 1). The ANOVA shows that all income groups up to EUR 22,500 present significant differences with income groups up to EUR 300,000. However, it should be noted that the effect size is smaller: *d_Cohen_* = 0.183 (Small Effect).

## 5. Discussion

The results obtained reveal some characteristics of the population with a greater risk of fear in situations that favour infection by SARS-CoV-2, which allows elaboration of a practical profile that can be used to develop of prevention strategies in the field of public and community health.

Owing to the large sample collected, which represents people of all age groups (over 18) and socioeconomic status, we were able to perform a predictive analysis that provides us with a clear view of what could happen in the event of a new outbreak.

In later stages of the pandemic, some studies were published in which the perception of fear was also measured by using validated tools. Even though the results cannot be compared with ours because of the differences in the time period studied and the type of population (university students [35] and people living in unwanted loneliness [36]), it is striking that in both papers there is a correlation between fear and perceived risks with beliefs about future states and uncertainty about health status. In Bottemanne and Friston’s study another variable was added: the outcomes of policy strategies, which can determine individual protective behaviours [37].

Similar studies have shown that a higher percentage of women respond to surveys [24,38,39], which was also true for this study. Being a woman seems to increase the fear of infection and risk perception, which has also been observed in other studies [40]. The fact that women answer this type of survey to a greater extent may give us additional information that supports the idea that they are the ones who show the most fear and concern in situations where there is a risk of infection [41].

Another significant factor is age: the older a person is, the more anxious they become. Taking into account that the risk of death from COVID-19 is higher in elderly people [42], and that the most vulnerable and most likely to die from this disease are people over 60 years of age and/or with concomitant chronic pathologies [43], it is understandable that we obtained these results. On the other hand, when one’s health condition is qualified as “average”, we see an increase in the fear of infection and the rise of the idea that this health condition could worsen during confinement. However, health visits for chronic diseases are declining. Authors such as Toniolo claim that the decline in visits for cardiovascular diseases is not because of a lower number of cases, but rather to the population’s fear of infection during hospital visits [22].

The probability of fear of infection increases if the person has been infected by COVID-19 or if a family member, friend or coworker has been infected (or has died) [44]. Therefore, people who live with others are more afraid than people who live alone [45]. Within the collective perception, the greater the general effect at the European level, the greater the fear in countries that have been most affected [40,46].

A person’s financial status and level of education have a significant impact on their fear; people with higher income and a higher level of education display less fear of infection [41]. Studies carried out in the United States [47] and in England [48] show how people with lower income levels and less education have had a higher rate of hospitalisation due to COVID-19.

Preventive behaviour is not always associated with a decrease in fear, although greater access to information does seem to increase it [26]. However, our study, which concurs with other reports [27,49], shows that people who take the most protective measures are the ones who are the most afraid. Fear brings about a rise in the use of prevention measures, such as hand washing, avoiding crowded places and contact with potentially contaminated surfaces [50]. Knowledge about the disease leads to preventive behaviour [50] and helps to diminish fear [46]. The adoption of preventive behaviours reduces the initial collective fear perception and has an impact on the progressive reduction of fear.

It is important to note that people with less education and the elderly have less access to reliable information and stay up to date with information gleaned from social media [6]. This paper shows that the most frequent sources of information are television, radio and written press, followed by official sources and scientific documents. Social media, in this case, would be in the last position. However, social media plays a greater role in studies such as that conducted by Kwok, in which 84% of respondents claimed to obtain information from social media despite the fact that only 26% considered it to be a reliable source [51]. Other authors [39] have discovered that the more frequently one uses social media, the greater the perceived risk of the COVID-19 virus. In our study, however, we found no association between fear and the source of information.

## 6. Limitations

Owing to the exceptional circumstances caused by the pandemic, snowball sampling was determined to be the most appropriate given the research limitations imposed by this situation. Although there was a high level of participation, this type of sampling does not allow the results to be extrapolated to the whole population, although it provides an idea of how fear influences behaviour in situations of crisis. Finally, it should be noted that the method chosen for distributing and collecting data imposes a limitation for a part of the population that does not have access to this technology.

At the time of the study, there were no validated tools to measure attitudes and fear related to COVID-19, so an ad hoc tool was developed to collect the variables of interest for the study. Specific scales were subsequently developed and validated, as in the Italian National Epidemiological Survey on COVID-19 (EPICOVID19). This self-administered, cross-sectional survey was used at national level to determine, among other aspects, fear of COVID-19. Furthermore, its items correspond to a large extent to those elaborated in our design, being the tool used in other different studies [52,53,54,55].

As we have explained, the fact that the sample is not random conditionally limits the scope of the results to the subgroups represented. However, given the large sample size, it is possible to assume a certain level of quantitative representativeness, with partially controlled biases. This has allowed us to draw provisional conclusions of interest, in view of the lack of knowledge during confinement when it was difficult to construct a suitable sample framework, which can guide future work while we await the opportunity to work with statistically representative samples.

## 7. Conclusions

There is a correlation between the profile of the people with the highest level or fear based on age and gender. Older people and women are more likely to be afraid of becoming infected by the disease. In addition, having someone infected with COVID-19 in the immediate surroundings, living with others and belonging to the most socioeconomically vulnerable group of people increases fear of infection.

Therefore, it seems appropriate that in order to reduce fear in the population, measures related to health literacy of the population should be implemented, thus converting fear into adequate and effective protective measures that reduce the risk of infection among the most vulnerable populations.

Effective communication is crucial to promoting collective action to prevent the spread of the virus. Health literacy, habits and social norms in different populations are central components of public health interventions.

Knowing and understanding people’s fears and behaviours towards COVID-19 can help health institutions to take measures to ensure preventive behaviour in this and future pandemics.

Health systems may in the future benefit from past pandemic experience in order to improve their community response to potential new pandemics.

While contact tracing and isolation are crucial components of intervention, privacy and human rights issues need to be considered.

Understanding community responses to containment policies will help end current and future pandemics around the world.

Fear perception in Spain (and its role in motivating preventive health behaviour) could help policy makers design evidence-based risk communication strategies.

This paper provides relevant information for the public health sector since it contributes first-hand knowledge of population data that is highly useful in terms of prevention. Understanding the experiences of people who have lived through this pandemic can help create more effective future intervention strategies and, consequently, improve planning and management strategies for crisis situations.

One of the strengths of our study is its early conduct, as it was carried out a few days after the WHO declared the COVID-19 outbreak a pandemic and the imposition of strict security measures in several European countries.

## Figures and Tables

**Figure 1 ijerph-19-00834-f001:**
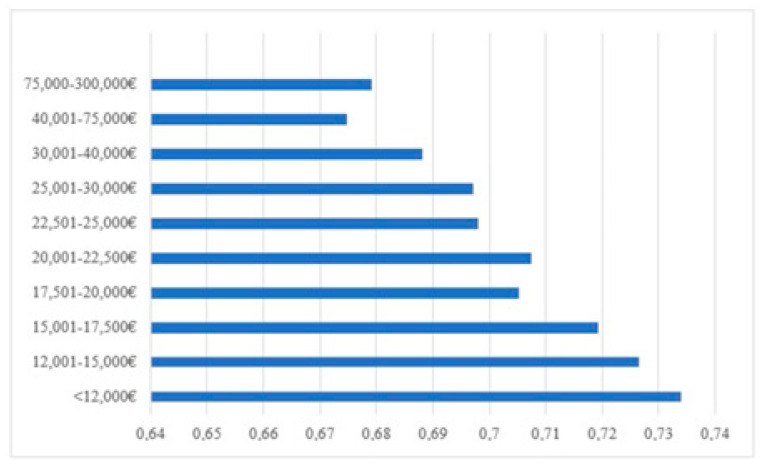
Distribution of annual salary and likelihood of fear of infection in the regions of Madrid, Cantabria and the Canary Islands.

**Table 1 ijerph-19-00834-t001:** Descriptive analysis of the study variables in the regions of Madrid, Cantabria and the Canary Islands.

	N: 16,201
	% (N)
Sociodemographic variables group	
Marital status	
Single	28.8 (4662)
Separated/Divorced	10.2 (1647)
Married/Partner	59.0 (9652)
Widowed	2.0 (330)
Level of education	
Primary	8.8 (1429)
Secondary	33.3 (5389)
University graduates	57.9 (9383)
Employment status	
Employed	87.2 (11,718)
Unemployed	12.8 (1724)
Active population	
Yes	83.0 (13,442)
No	17.0 (2752)
Perception of family financial situation	
Good	44.7 (7244)
Average	42.4 (6871)
Poor	12.9 (2086)
Cohabitation	
Alone	10.8 (1756)
Accompanied	89.2 (14,445)
Group of variables: health factors and risk perception	
Health status during confinement	
Good	74.8 (12,112)
Average	22.0 (3569)
Poor	3.1 (509)
Infection in the home	
No	96.2 (15,591)
Yes	3.8 (610)
Healthcare worker family member	
No	75.8 (12,281)
Yes	24.2 (3920)
Fear of infection	
No	28.9 (4674)
Yes	71.1 (11,527)
Fear of a family member becoming infected	
No	3.8 (611)
Yes	96.2 (15,590)
Protective measures used	
Adequate	72.4 (11,723)
Inadequate	27.6 (4478)
Means of obtaining information	
Press/Radio/Television	66.6 (10,792)
Social media	9.5 (1540)
Official media/scientific documents	23.9 (3869)
Source: Compiled by authors	

**Table 2 ijerph-19-00834-t002:** Fear of infection according to sociodemographic variables.

	NO	YES	Pearson Chi-Square	Asymp. Sig. (2-Sided)	Cohen *d*	OR
Age						
=30 years (ref.)	38.5% (714)	61.5% (1141)	111.03	0.000 **	0.1661	
31–64 years	28.1% (3675)	71.9% (9415)				1.603
=65 years	22.7% (285)	77.3% (971)				2.132
Gender						
Male (ref.)	34.0% (1591)	66.0% (3089)	84.89	0.000 **	0.1452	1.430
Female	26.8% (3083)	73.2% (8438)				
Marital status						
Single (ref.)	33.5% (1560)	66.5% (3102)	84.73	0.000 **	0.1450	
Separated/divorced	31.1% (513)	68.9% (1134)				1.112 *
Married/Partner	26.4% (2521)	73.6% (7041)				1.405
Widowed	24.2% (80)	75.8% (250)				1.572
Level of education						
Primary	18.3% (261)	81.7% (1168)	109.57	0.000 **	0.165	1.991
Secondary	27.5% (1480)	72.5% (3909)				1.164
University graduates (ref.)	31.3% (2933)	68.7% (6450)				
Employment status						
Employed (ref.)	29.7% (3476)	70.3% (8242)	5.25	0.022 *	0.039	1.142
Unemployed	27.0% (465)	73.0% (1259)				
Perception of family financial situation						
Good (ref.)	31.3% (2264)	68.7% (4980)	39.13	0.000 **	0.098	
Average	27.3% (1876)	72.7% (4995)				1.132
Poor	25.6% (534)	74.4% (1552)				1.201
Cohabitation						
Alone (ref.)	34.6% (607)	65.4% (1149)	31.36	0.000 **	0.088	1.401
Accompanied	28.2% (4067)	71.8% (10378)				
Health status during confinement						
Good (ref.)	31.1% (3768)	68.9% (8344)	118.43	0.000 **	0.172	
Average	22.1% (787)	77.9% (2782)				1.436
Poor	23.2% (118)	76.8% (391)				1.132
Protective measures used						
Adequate	24.2% (2836)	75.8% (8887)	448.36	0.000 **	0.337	1.95
Inadequate (ref.)	41.0% (1838)	59.0% (2640)				
Healthcare worker family member						
No	28.7% (3519)	71.3% (8762)	0.950	0.330		
Yes	29.5% (1155)	70.5% (2765)				

Source: Compiled by the authors. (*) Single versus separated/divorced persons have a nonsignificant OR. All other ORs are significant with an NC 95%. The reference category (ref.) is noted in order to interpret the ORs. ** *p* < 0.001; * *p* < 0.05.

**Table 3 ijerph-19-00834-t003:** Variables in the equation used in the regression analysis (likelihood of fear).

	B	S.E.	Wald	df	Sig.	Exp(B)	95% C.I. for EXP(B)	
							Lower	Upper
Gender: Woman (Sex)	0.360	0.042	71.63	1	0.000 **	1.43	1.32	1.56
University Studies (ref.)								
Primary Studies (PS)	0.691	0.084	67.21	1	0.000 **	1.99	1.70	2.35
Secondary Studies (SS)	0.148	0.044	11.36	1	0.000 **	1.16	1.06	1.26
COVID-19 Infection	0.293	0.071	17.09	1	0.000 **	1.34	1.17	1.54
Family Finances Good (ref.)								
Family Finances Average (Ffa)	0.124	0.042	8.55	1	0.003 **	1.13	1.04	1.23
Family Finances Bad (Ffb)	0.179	0.068	6.94	1	0.008 **	1.20	1.05	1.37
Cohabitation: Accompanied (Co)	0.332	0.060	30.46	1	0.000 **	1.40	1.24	1.57
Current Health Good (ref.)								
Current Health Average (Cha)	0.359	0.051	48.72	1	0.000 **	1.43	1.29	1.58
Current Health Poor	0.118	0.119	0.98	1	0.322	1.13	0.89	1.42
Age (Age)	0.013	0.002	71.02	1	0.000 **	1.01	1.01	1.02
Adequate protection when leaving home (AP)	0.666	0.041	260.28	1	0.000 **	1.95	1.79	2.11
Constant	−1.07	0.111	94.04	1	0.000 **	0.34		

Source: compiled by the authors. Acronyms for the expression of the regression model estimation: Sex = Gender; PS= Primary Studies; SS = Secondary Studies; COVID = COVID-19 Infection; Ffa = Family Finances Average; Ffb = Family Finances Bad; Co = Cohabitation: Accompanied; Cha = Current Health Average; Age = Age; AP = Adequate protection when leaving home. ** *p* < 0.001.

**Table 4 ijerph-19-00834-t004:** Comparison of means/ANOVA for certain subgroups regarding the probability of fear of infection.

	N	Mean	Std. Dev.	Std. Error	T/F	Sig. (2-Tailed)
Sex						
Male	3878	0.654	0.108	0.002		
Female	9908	0.728	0.091	0.001	−37.545	0.000 **
Education						
Primary	1167	0.824	0.066	0.002	1215.28	0.000 **
Secondary	4443	0.721	0.093	0.001		
University	8176	0.683	0.098	0.001		
Marital status						
Single	3899	0.666	0.108	0.002	329.61	0.000 **
Separated/Divorced	1375	0.725	0.097	0.003		
Married/Partner	8230	0.722	0.094	0.001		
Widowed	282	0.754	0.091	0.005		
Active population						
Yes	10,084	0.698	0.101	0.001		
No	1377	0.725	0.102	0.003	−9.216	0.000 **
Live alone?						
Alone	1476	0.645	0.115	0.003		
Accompanied	12,310	0.715	0.098	0.001	−22.427	0.000 **
Protection measures						
Adequate	9993	0.753	0.065	0.001		
Inadequate	3793	0.588	0.084	0.001	122.48	0.000 **
Family financial situation						
Good	6415	0.684	0.101	0.001	355.59	0.000 **
Average	5748	0.722	0.099	0.001		
Bad	1623	0.746	0.098	0.002		
Health condition						
Good	10,528	0.686	0.099	0.001	1123.0	0.000 **
Average	2853	0.779	0.078	0.001		
Bad	405	0.748	0.086	0.004		
Healthcare worker family member						
No	10,374	0.708	0.080	0.007	1.930	0.054
Yes	3412	0.705	0.079	0.001		
Infection in the home						
No	13,269	0.706	0.080	0.001	−7.020	0.000 **
Yes (Test+)	517	0.731	0.078	0.003		
Fear of a family member becoming infected						
No	526	0.689	0.075	0.003	−5.481	0.000 **
Yes	13,260	0.708	0.080	0.001		
Fear of infection						
Low	6032	0.691	0.105	0.001	178.25	0.000 **
Medium	4714	0.712	0.099	0.001		
High	3040	0.732	0.094	0.002		
Means of obtaining information						
Press/Radio/Television	9266	0.709	0.103	0.001	2.261	0.104
Social media	1301	0.706	0.105	0.003		
Official media/scientific documents	3219	0.704	0.098	0.002		

Source: compiled by the authors. Note: Student T-tests for comparison of means for independent samples in the case of two groups; F test for ANOVA when there are more than two groups. All tests were performed for an NC95%. ** *p* < 0.001.

## Data Availability

Data supporting the findings of this study are available upon request from the reference author (Ana María Recio Vivas). The data are not publicly available as they contain information that could compromise the privacy/consent of research participants.

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
