# Peer review of "Fear and Attitude towards SARS-CoV-2 (COVID-19) Infection in Spanish Population during the Period of Confinement"

_ijerph, 2022, doi:10.3390/ijerph19020834_

Round 1
Reviewer 1 Report
I would like to thank you for the opportunity to review this manuscript. The paper aims to investigate and analyze the fear of infection from Covid-19 among the Spanish population during the state of emergency. I appreciate the importance of this work, particularly as it captures the characteristics of the country through the selection of three types of population. I am glad the authors brought this to the paper. I do have some suggestions to improve the validity and contribution of the paper, which I make below.
Introduction: The introduction is full of information and the connection among each paragraph may appear unclear at first reading. Certainly, the authors have done a good job of background analysis. Despite this, however, some improvements can be made.
#1 I would suggest that the authors use subsections to better clarify the logic behind the introduction (e.g., the pandemic situation, the fear, the consequences, and a short paragraph with aims); this will also allow to highlight the variables that are important to understanding the paper (i.e., fear of covid).
#2 I would also suggest adding some references when referring to other studies (i.e., p. 3, lines 98-103), (e.g., a review from Lausi et al., 2021, Burrai et al., 2020, Quaglieri et al., 2021, Mari et al., 2020/2021) and when referring to compliance with rules (e.g., Cava et al., 2005, Tomczyk et al., 2020) to make the paper more accountable.
Method:
Study design and participant selection: I found this paragraph well written and clear; I however have some questions:
#1 Was the questionnaire in Spanish? How were any non-native speaking participants identified? Were they included in the analyses? The authors should clarify to identify any biases in data collecting.
Tool and study variables: I found this paragraph clear and concise.
#2 I would suggest moving the information about the analysis in the following paragraph (e.g., p. 4, line 162).
Analysis
#3 Please, add a reference for the software.
Results:
#1 I would recommend highlighting with asterisks the significant results in each table.
Overall, the results are well written and clear to follow.
Discussion: The discussion section summarizes the findings and well-links them to the introduction, with a good job of elaborating and citing references; I would suggest adding some information about the novelty of the study and the practical implications
I congratulate the authors for the good job done. I hope that my comments are constructive and helpful and I hope to see this paper published.
Author Response
Thank you very much for your input. We have made a number of changes to the article based on your comments as we considered them appropriate, and they have contributed to improving the quality of the document. We respond to your contribution in the attached document. Modifications to the text are shown in yellow.

Reviewer 2 Report
This is an interesting account of a large scale survey with convenience sample, which identifies several ‘risk factors’ for fear of COVID-19 in a Spanish population, and discusses links between fear and preventive behaviours. The sample size is impressive and the findings are useful for understanding different demographics’ likelihood of reporting feeling fear of COVID-19 infection. The measures are quite basic, e.g. fear of infection is evaluated with a simple binary, ‘yes’/’no’ question, and I think more acknowledgement of the potential limitations of this is needed. There are some extra details needed about the methodology and analysis, in order to help the reader fully understand how some of the variables were measured and analysed.
Some specific, section by section comments follow below:
Abstract
I recommend stating whether the associations between fear of infection and the other variables listed in lines 23-25 were statistically significant.
Introduction
Please clarify the sentence ‘Risk perception is associated with increased preventive behaviour’. Is this a general statement about proven associations in the existing literature, or is it a primary finding from this study? If the latter, please add some detail to indicate what measure of ‘preventive behaviour’ was used, and note whether this was a statistically significant correlation.
Lines 58-59, where you state ‘these measures have been endorsed by the population as shown in the McFadden study’, it would be helpful to flag what the McFadden study involved; which population and what do you mean by endorsed? Complied with?
Lines 62-63, repetition of ‘to go to work’ is a little confusing – if everyone was allowed to go to work is it necessary to repeat that essential workers could leave to go to work?
There is a large section from lines 69-111 where the authors discuss how fear can be either detrimental or adaptive, depending on its proportionality, etc. This tends to prime the reader to expect the study to try and pin down what degree of fear is associated with positive adaptive responses vs. negative ‘collective fear’, harmful responses like resource hoarding. I think this section would benefit from reframing slightly to make it clear that the main focus of interest in this study (beyond identifying which demographics are most fearful) is the relationship between fear and preventive behaviours.
Line 71, the wording ‘the worst disease of all’ is a little controversial to describe collective fear – personally I would recommend a milder phrasing.
Line 120, I think you need to specify what you are referring to predictors of.
The last paragraph of the introduction speaks about the importance of finding out what people are afraid of, implying that qualitative understanding of what types of fears people have is part of the study. However, the survey does not seem to delve into what it is they are afraid of, other than very broadly asking whether or not they are afraid of infection with covid-19; I think this paragraph needs editing to avoid giving the impression that the study will be looking into the nature of the fear in more depth.
Method
Lines 151-152 I think a little more information is needed about how people were recruited through ‘social media and Scientific Societies, as well as healthcare institution’. Was this through groups or private networks of the researchers? Through emails to mailing lists? Etc.
Line 168, please clarify what is meant by ‘active population’.
Line 171, a little more explanation is needed of how you obtained data on ‘means of obtaining information and protection measures used’. E.g. was this multiple choice? Free text? How did you score it?
Line 178 please clarify what the continuous variable ‘infection’ consisted of, i.e. what were participants asked and what responses did continuous scores represent?
Line 194, please explain more about the ‘new variable with the probabilities of risk’ and how this was derived from the analysis. This is returned to in lines 252-295 where a series of ANOVAs and t-tests are done. I may be missing something, but I am struggling to see how this adds value beyond the data already presented, and how the (presumably continuous?) risk probability score is arrived at, given that the participants seem to have only answered one binary question about this.
Discussion
‘Risk perception’ is discussed as being associated with more preventive measures; it was not clear from the method/results whether/how risk perception was measured, so I wonder if ‘risk perception’ is being used as a synonym for ‘fear of infection’. This seems fairly logical, but still, I think the authors need to be explicit and include a justification for inferring that their binary measure of ‘fear of infection’ is a proxy for ‘risk perception’.
Line 346-347, it would be worth expanding on how you reconcile the idea that preventive behaviours can diminish fear with the finding in this study that fear and preventive behaviour are positively correlated. Is the hypothesis that the preventive behaviours will eventually reduce fear over time, perhaps?
Lines 350-357, discussion of the findings regarding source of information and fear is tricky to follow without knowing how you measured, scored and analysed impact of source of information (see earlier comment).
Lines 363-364, more justification of how that conclusion is reached from the data (or more cautious wording) is needed (e.g. ‘it may be the case that…’). It seems like some assumptions are being made, as I don’t think health literacy was specifically assessed.
Section 6 is supposed to be the ‘limitations’, but reads more like a summary/concluding paragraph. This section should spell out the limitations of the study.
Author Response

(The authors gave the same response as above.)

Reviewer 3 Report
This paper is focused on the analysis of risk factors associated with fear of SARS-CoV-2 infection from the 16th to the 21st of April 2020 during the period of mandatory confinement in three different regions in Spain. In particular, the Authors used a self-administered online questionnaires asking for several social, economic and health associated characteristics. The study population was large and selection bias appear to be sufficiently considered and contextualized in the Methods.
The social issue of this work appears to be quite relevant at present and in the future management of communications and decisions concerning pandemics. The theoretical basis is sufficiently displayed in the Introduction. Description of statistical methods is quite exhaustive. The methodological approach of this work is consistent and adequate for the aim of the study. Qualitative and quantitative findings are adequately summarized in “Results” even if some minor English revisions are required to better present them. The Discussion is coherent with main results. The impact of the findings may acquire relevance due to large number of administered questionnaire. To note that Authors used “Limitation” section to present the conclusions of their work rather than outline limitations.
Author Response
Thank you very much for your input. We have made a number of changes to the article based on your comments as we considered them appropriate, and they have contributed to improving the quality of the document. We respond to your contribution in the attached document. Modifications to the text are shown in yellow.

This manuscript is a resubmission of an earlier submission. The following is a list of the peer review reports and author responses from that submission.
Round 1
Reviewer 1 Report
Based on the current condition of the designated manuscript, I cannot justify its recommendation for publication. The convoluted nature of the writing style and science involved, and the prevalence of missing information (e.g. what is the real extension of the topic?) requires that I recommend rejecting the manuscript from publication, with optional resubmission. In my opinion that the current condition of the manuscript is not acceptable for publication in IJERPH
Minor points
-Table 1 need to be redone and presented in a better way
-The expression of the regression model is not clear not to be redone
-on what bases are the three regions chosen?
-Similarity index need to be reduced, it is 31% not, better to bring down less than 15%
Reviewer 2 Report
Paper that addresses a topic of real interest, not only for this pandemic, but for future pandemics that are sure to follow.
The methodology is correct, the discussion is correct
The authors should vastly improve the introduction. In the results, the tables are not well formatted. The conclusions do not correspond with rigour to this work.
El abstract is correct
The introduction and justification do not define the main term of paper:the fear of infection from COVID-19
The authors explain the scenario of the pandemic. Their argument is focused towards justifying why there might be fear.But they do not describe how fear is built, why it arises in such a situation.Although I have not written about fear in the COVID-19 pandemic, there is previous literature on fear in collective processes.The authors will have to improve this introduction in this regard.
Table 1 cannot be read. The formatting of the table needs to be improved.
All tables should be improved
Figure 1 should be improved
The conclusions are beyond the scope of this paper.
For example:
In order for people to stay at home and at the same time maintain their livelihoods, 380 economic and social support is essential.
Understanding community responses to containment policies will help end current 386 and future pandemics around the world.
Authors should limit the conclusions to what they have demonstrated with their paper.